# A Preliminary Contribution towards a Risk-Based Model for Flood Management Planning Using BIM: A Case Study of Lisbon

**DOI:** 10.3390/s22197456

**Published:** 2022-10-01

**Authors:** Graziella Del Duca, Gustavo Rocha, Marta Orszt, Luis Mateus

**Affiliations:** 1Centro de Investigação em Arquitetura, Urbanismo e Design (CIAUD), Faculdade de Arquitetura, Universidade de Lisboa, 1349-063 Lisbon, Portugal; 2Faculdade de Arquitetura, Universidade de Lisboa, 1349-063 Lisbon, Portugal

**Keywords:** CIM city information modeling, 3D BIM map, BIM flood risk-based urban planning, large BIM file, flood hazard, SLR sea level rise, coastal flood management, BIM urban plan, flood risk exposure, flood risk vulnerability

## Abstract

Preparing a city for the impact of global warming is becoming of major importance. Adopting climate-proof policies and strategies in response to climate change has become a fundamental element for city planning. To this end, this research considers a multidisciplinary approach, at the local scale, able to connect urban planning and architecture, as a vital base for considering a coastal cities’ ability to control the consequences of climate change, specifically floods. So far, there is a scarcity of research connecting sea ground and land surveys, and this study could become a foundational reference for coastline settlement management using BIM. We found in BIM (Building Information Modeling) a possible tool for managing coastal risk, since it can combine crowdsourced data for geometric and information modeling of the city. The proposed BIM model includes a topography used for 3D thematic maps, a riverbed model, and a waterway model. This model aims to facilitate coordination across separate actors and interests since the urban area model is always updatable and improvable. Focusing on a case study of Lisbon, we developed risk-based 3D maps of the area close to the shoreline of the Tagus River.

## 1. Introduction

Areas near water have multidimensional potential and an important role in human economic [1], social, and cultural development. These areas are the most sensitive to consequences of climate change. They are impacted by inland and coastal flooding, storms and cyclones, which pose a real threat to communities, economy, and cultural heritage. Major cities and world capitals are historically situated by the seacoast and rivers. Although catastrophic events reported in the newspapers point out the high risk to which they are subjected with increasing frequency, there is still a lack of implementation of climate change-adaptation policies [2,3] Preparing coastal areas and cities for threats of climate-related hazards is undeniably important, especially for those vulnerable human and natural systems that lack the capacity to cope and adapt to upcoming changes. Damage in cities near water may reach hundreds of billions of US dollars worldwide if the high-emission “RCP8.5” warming scenario [4] becomes a reality. Nevertheless, only the most aggressive mitigation scenarios (1.5–2.0 °C) can slow the acceleration of sea-level rise (SLR). A 3 °C Global Warming Level would already result in twice as many people affected by floods [5]. Vast areas and their inhabitants would be affected by sea-level rise (SLR) [6] even with applied climate change mitigation [7]. Globally, the area of land less than 2 m above mean sea level is 649,000 km^2^ [8]. By the end of the 21st century, coastal flood damage has been forecasted to increase at least 10-fold [9].

Because of its extensive consequences, coastal flood risk needs to be seen in a wide scope of urban planning. Holistic and sustainable actions at urban scales are possible if a vast amount of data is accessible from one source, allowing a view into the topic from the general to the particular, with the possibility of choosing the level of detail matching a chosen case. One of the most suitable solutions would be to use systems in City Information Modelling (CIM), which are analogous to Building Information Modelling BIM [10].

CIM provides comprehensive information to people managing cities in the form of an urban database that can be constantly detailed and updated, and extends beyond data provided only by geographic information systems (GIS) [11] or simple CAD. It includes spatial data, and also economic, transportation, underground network [12], geographical and architectural data, connecting GIS, BIM and the data of smart cities [12]. Using CIM would be especially useful, given that the 2014/24 European directive in the construction field aims at hitting the target of having BIM [13] as a standard for all publicly funded construction works by 2025. ISO 37120:2018—Sustainable development of communities; Indicators for City Services and Quality of Life [13]—includes standardized indicators in subjects as transportation, urban planning, recreation, health, and shelter. The aim is to make their management more coherent and efficient, since it is both important and complex. CIM allows a unification of multidisciplinary and multi-dimensional [12] spatial data, which can take care of a vast variety of themes stated in ISO 37120, from which results prove possible automation of 53 indicators out of 100 proposed by the International ISO 37,120 Standard [14]. Therefore, CIM is an efficient way of management and simulation of a sustainable urban built environment, if applied in practice [15].

Existing research shows the benefits of integrating BIM and GIS in historic areas [16,17] for facilitating their maintenance and conservation. Merging GIS and BIM allows visualization how city elements interact with one another. Research has already demonstrated that flood damage can be assessed and visualised with BIM/GIS integration [18] when applied to a specific building or a small group of buildings. Unfortunately, the coordination between BIM and GIS still lacks a unified standard for information exchange [19,20]. To raise their interoperability, protocols on data management are needed, from data acquisition to data delivery. Standardization of processes would help architects and urbanists to inter-operatively provide the adequate level of detail and development, and governors would be able to execute information deliveries, building uniform databases and datasets, without ambiguity or clash. Great standardization is already employed in BIM due to open international industry foundation classes, data dictionaries, and data delivery manuals. Nowadays, even though BIM and GIS highly correlate in the scope of urban planning, they include parts of one another rather than creating a new, unique system of data management [21].

So far, most of the detailed research on the topic of BIM for flood management and city planning considers the relationship between single buildings [14], public space, and water, rather than complex urban settlements. BIM, in these cases, is usually integrated with a plug-in or external software for flood-simulation [14]. Further, there is fragmentation in the instruments and management bodies for riverine areas close to historical centres. For instance, riverside areas in the proximity of city centres tend to be viewed through the lens of historical value and ongoing urban renewal [22,23], disconnected from the subject of climate change threats.

Contrasting with wide research on climate change, the use of new technologies and innovative methodologies to improve management and urban planning is scarce. The use of software allows microclimate simulation [24] and is dissociated from considerations about aspects such as the historic and social value of heritage in the area of analysis. Additionally, geodetic data are usually separated into bathymetric or terrestrial studies, rarely revealing results on the sea/land connection. The creation of a tool that permits the use of different sources and types of data should be prioritised in order to generate an inclusive and sustainable urban governance.

The aim of the present proposal was to address the disconnections described above to merge topics related to climate, heritage, and urban planning, allowing for the addition of crowd-sourced data to enrich and safeguard human settlements. In this research, we exploited BIM as main tool for flood risk management of city buildings, using BIM internal features only (instead of the traditional CAD and GIS). This involved working with a large project area. We applied visual language to speed up repetitive and time-consuming process. We tested our procedure in a chosen exemplary area, Lisbon. The application of the methodology to the case study provided us the opportunity to verify how to apply the theorised contents on flood risk management to a BIM model. The experiment provided us information on the dynamics of file growth, in that it is imperative to pay attention to the size of the project file, working at this scale, as a too large file would be unmanageable. 

The remainder of this study is organized as follows: in Section 2, we introduce the proposed method, while the results of the model are provided in Section 3. Section 4 presents our conclusions and suggests some possible avenues for future research.

## 2. Materials and Methods

### 2.1. Introduction

The risks connected to flooding can be reduced either by structural interventions such as embankments, retention basins, spillway drains, or by non-structural interventions, such as risk-based city management, or emergency management. In 2007, the European Flood Directive (EFD) [25] legislation went into effect, requiring Member States to assess and identify areas at risk. The EFD aims “to reduce the risk of adverse consequences from flooding, especially for human health and life, the environment, cultural heritage, and economic activity”. However, according to articles 5 and 6, the EFD mandates mapping only for those zones exposed to risk or possibly exposed to risk and does not require a quantitative risk estimation [26]. The type of analysis, the scale of representation and the tool to use for risk assessment, as well as risk modelling, are not specified in the EDF.

We adopted a three-step procedure for the type of risk assessment suggested by the European directive. Starting from assessing flood risk factors, we identified the most exposed areas, defined possible hazards, and considered possible countermeasures. In addition, we introduced some markers that would effectively improve flood risk management and operational directions for the city, such as “building value” and “priority level”. The factors considered for the risk assessment were adapted and applied to a BIM tool.

The details provided here are intended as a guide only. This article is not intended to provide technical specifications. The hazard and control procedures listed may not be a comprehensive list as per the analysis, evaluation and mitigation solutions. It must be ensured that the proposed contents can provide an acceptable control for the significant hazards that are present in a particular circumstance, they should be should integrated or amended according to local regulations. If doubts, queries or concerns arise, relevant regulations should be referred to and professional advice sought.

#### 2.1.1. Type of Analysis

The types of analyses that are usually conducted on natural hazard risk assessments can be classified either as non-probabilistic (NP) or probabilistic (P). Probabilistic refers to studies that assess expected annual impacts by integrating across return periods based on a probabilistic stochastic event set [27]. Studies on floods use both methods even if, for more in-depth analysis, there has been a move towards more probabilistic studies in recent years [27]. Not all cities can cope with a probabilistic analysis, as adequate records on floods in past years are often scarce or non-existent [28], or simply because the approach requires more time and money to be implemented. The analysis carried out for this research are based on a non-probabilistic (NP) evaluation. The choice of using NM analysis lies in the fact that we wished to use input data that is easy to find and access. Our research required digital cartographies that are commonly available on public online databases of city departments for land and urban management. Files can be provided for free or for a small fee.

According to the European commission document titled, “Floods Directive Reporting Schemas” [25], the methods for reporting are different, and quantitative estimation is optional. The description of the risk and its factors can be indicated as a range, as a percentage of the total GDP for the flood event, by classes (Insignificant, Low, Medium, High, Very High), or by means of other numerical measure indicative of the degree of potential adverse consequences, leaving a wide choice for implementation [26]. We grouped buildings by classes and assigned values from 1 to 5 (1 low, 5 high) for each analysed risk element.

The process we used for risk assessment adopted the same conceptualisation as the Sendai Framework for Disaster Risk Reduction [29], for which risk is the product of the elements: exposure, vulnerability, and hazard. Exposure refers to the location of economic assets or people, vulnerability refers to the susceptibility of those assets or people to suffer damage and loss (e.g., due to unsafe housing and living conditions, or lack of early warning procedures), and hazard refers to the hazardous phenomena itself [27].

The next section describes our consideration of Exposure, Vulnerability, and Hazard for the implementation of a risk-based model for flood management planning using BIM.

#### 2.1.2. Exposure

Risk exposure is the likelihood that a risk will occur. Since we did not use the so-called “flood depth map”, which details a probabilistic analysis that shows the area which could get flooded if the water level rises to a particular elevation in a certain period time, we had to combine other parameters for identifying the areas at higher risk and those considered safe. Altimetric values and the distance from the water—estuary, riverside, or seacoast—defined our reference levels of risk exposure for the city assets. For low altitudes and major proximity to the water we defined the highest levels of exposure connected to the flood risk. The 5 risk exposure rating levels used to determine the likelihood of a flood risk are described as follows:E1: Highly UnlikelyE2: UnlikelyE3: PossibleE4: Likely. A likely risk has a 61–90 percent chance of occurringE5: Highly Likely. Risks in the highly likely category are almost certain to occur

Attribution of the level of risk exposure to an area is defined by the reference values of risk exposure shown in the matrix below (Figure 1) that combine reference levels of elevation with the distance from the shore.

The matrix shows that the areas most likely to be involved in a flood are those closer to the water (from 0 to 400-m distance from the shore) for elevation values greater than 0 m and less than 4 m.

For lower altitudes, we also expect a (potential) higher level of hazard exposure, and a bigger economic impact on people and assets. Exposure is a necessary, but not a sufficient, determinant of risk. The second factor that defines risk is hazard severity, and it is intrinsic to level of vulnerability.

#### 2.1.3. Vulnerability

Vulnerability refers to the susceptibility of those assets or people exposed to risk to suffer damage and loss. It is possible for a building to be exposed but not vulnerable; however, to be vulnerable to an extreme event it is necessary to be exposed [30]. Vulnerability depends on several factors (proximity to coastline, presence of physical barriers, soil characteristics, building construction quality, materials and typology, building structure efficiency, and building reaction to previous floodings, among others). In most studies where vulnerability was considered, it was considered in a static scenario [27]. We focused this study on the fact that vulnerability can change over the time, and the aim of risk assessment is to improve flood preparedness of urban areas.

For flood vulnerability classification, we followed the detailed list proposed by the UK Department for Levelling Up, Housing and Communities for the National Planning Policy Framework, in Annex 3 [31]. The UK flood risk vulnerability classification groups assets according to the parameters listed below, and we attributed a risk vulnerability rating (from 1 low, to 5 high) to each group:V1: Water-compatible developmentV2: Less vulnerableV3: More vulnerableV4: Highly vulnerableV5: Essential infrastructure

The “essential infrastructure” category includes the infrastructure—transport for mass evacuation routes, structure for electricity supply, power stations, and water treatment stations—that need to remain operational in times of flood.

All the other assets are listed by increasing degree of vulnerability, from “water-compatible” to “highly vulnerable”. They follow an order based on their role in emergency operations, their structural capability to resist floods, and the amount of people that could possibly be affected during a flood. Buildings that combine a mixture of uses should be placed into the higher of the relevant classes of flood risk sensitivity.

#### 2.1.4. Hazard

A hazard is any source of potential damage, harm or adverse health effects to something or someone. The hazard evaluation of a flood event using large-scale analysis is often considered to be the probability that the flood itself would happen. In many flood risk analyses, the hazard value is normalised to 1.

However, we believe that it is important on this occasion to outline the main types of damage that a city, its inhabitants, and the economy, can suffer in case of a flood. Knowing these helps in choosing the most appropriate corrective action. We list the possible hazards related to flooding in a city in Figure 2.

### 2.2. Risk Management

#### 2.2.1. Risk Identification

In Figure 2**,** we list the possible risks connected to a flood hazard. To each type of risk, we give a value of exposure (E) and vulnerability (V) from 1 to 5 (1 = low, 5 = high). As stated in the previous paragraph, hazard (H) was given a default value of 1, so the risk formula that is usually expressed as:R = E × V × H(1)
can be rewritten as this second formula:R = E × V(2)

The maximum value for risk is 25. To each type of risk, we applied a mitigation action to reduce its severity. For some types of hazards, for example the flood itself, the exposure value cannot be changed since it is strictly dependent on altitude and distance from the waterline. We consider altitude and distance from the waterline invariable—unless embankments, dams, locks, or other similar works are carried out in the vicinity of the analysis area. For other actions of flood risk mitigation, when the hazard is restricted to a small area, the exposure to risk can be reduced by taking safety procedures. Generally, the most effective flood risk-mitigation actions are those that reduce vulnerability levels.

#### 2.2.2. Safety Risk Acceptance

Comparing the initial value of R*_t_* and the final value of R*_i_* in the second matrix of Figure 2, we can verify if a mitigating action may possibly reduce the hazard, and consequentially save lives and goods. We list the most common mitigations in the “Control for risk mitigation” column of Figure 2. However, risk mitigation is not always needed. The safety risk acceptance indicates for which values of R the risk is acceptable (1–5), tolerable (6–12), or unacceptable (13–25). The combined results of the risk exposure and vulnerability are presented in the safety risk assessment matrix (Figure 3).

#### 2.2.3. Resolution of Risk

Once the categories of risks have been profiled, they can be ranked into an ordered list representing the various threats that the authorities and professionals have to deal with in recovery projects. The more significant categories can then be examined and assigned with a high priority action by the project team. The maps of risk resolution initially only involved the effect of the floods on the population, then slowly included economic damage indices and land use. In the last few decades, we have seen a transition from addressing affected people only, to moving towards a broader range of risk indicators, including direct damage and, in the case of Dottori et al. [32], also indirect damage.

The resolution risk starts from vulnerability reduction and exposure reduction (when possible), which are the core elements of adaptation and disaster risk management. Actions on vulnerability have more effective results but require in-depth studies, and thus constitute an important common ground between the two areas of policy and practice [30].

As previously stated, possible actions that can decrease risk are described in Figure 2 in the risk mitigation column. Usually, the risk assessment ends at this point, possibly with maps that synthesize the risk at different scenarios (different water levels). However, we would like to emphasise the importance of further aspects—for post-flood damage management—that risk management plans should be taken into consideration as described in the next paragraphs.

#### 2.2.4. Risk Prioritization

Risk prioritization is the process of identifying all the risks and then deciding which are the most severe so these can be addressed first. From the safety risk acceptance matrix, we set the range of those risks that are unacceptable and those that need a mitigation action to be corrected and made acceptable. We grouped the actions of risk mitigation into two subcategories: those to be executed before a flood and those after. Risk evaluation is useful only if preventive actions for flood management, and for countermeasure financing, are set and executed before catastrophic events. We also believe that evaluations of the importance of built, natural and intangible heritage, are equally needed before any flood occurs. These represent the framework within which to develop a prioritisation plan to safeguard the most significant assets after a flood occurs. An assessment of the priority to be given to city buildings would speed-up post-flood restoration and reconstruction.

Priority classes can be defined according to the needs of the city and its community since they take into account a list value and “ad hoc” considerations that depend on social identity and historic-artistic background. The evaluation of the value of local assets, and the consequent definition of the priority class, should take into account the principle of sustainable future. Once the indicators that set the classes for value and priority are defined, the results can be incorporated into maps.

### 2.3. City Information Modeling

#### 2.3.1. Interoperability and Resolution

The two structures on which the proposed CIM model is based are the city’s “risk information” modelling and the city’s “morphology” modelling. One contains information and the other contains geometric elements that display the existing state of the city. Both are essential. The geometric model should respect pre-defined characteristics of accuracy and detail, and the information should respect certain requirements, namely, BIM standards for name codes, exchange formats, and delivery specifications. A lack of accuracy of the geometric model can cause errors in the information contained in the model.

A lack of standardization in the information exchange requirements can create ambiguities, a need for rework, lack of interoperation and inconsistency. Since BIM and CIM are time-consuming operations, it is important to set the models correctly from the very beginning to avoid rework. If the process we describe is to be used for city management, we expect it to be used in a federated model system. For this reason, we have to ensure interoperability between stakeholders.

Starting from setting up the model, we used a dataset registered on GPS coordinates. This allows for a shared coordinate system, as required for all BIM models with a maturity level superior to 2, so that other files—CAD or other 3D models—can be linked in the BIM project.

#### 2.3.2. Input Dataset and Model of the Existing State

A geometric model of the city is the infrastructure upon which to build the information we require the model to contain. Both the model and its information must be flexible and improvable. A level of detail or development higher than the scope of the project can create inconsistency with other linked models or generate over-constraints that make the model hard to integrate with extra information, or simply difficult to manage by other stakeholders. Since we worked at an urban scale, we wanted to provide the model with enough detail to be reusable for other purposes. The objects the model should contain should be only those needed to finalize all the process of risk assessment described in the previous paragraphs. For this reason, we required our model to incorporate:The Digital Terrain Model (DTM) of the area we wished to analyse. The DTM should represent the bare ground surface excluding trees, buildings, and any other objects on the ground surface.A 3D model of the nearest water body.

The latter element is used only to measure building distances and location heights to the waterline, as values needed to define the risk exposure.

The DTM was generated using the BIM object “topography,” which is the base layer on which visualized the maps produced during the risk assessment. Topographies in BIM are usually generated automatically, from an input file containing terrain-level information, such as dense point clouds (acquired from terrestrial laser scanning or built by structure-by-motion process), sparse point clouds (GPS points or topographic points), or 3D topographic CAD drawings.

Any of the data listed above are valid inputs for analysis if the density of the data is enough to produce a model of the terrain of the city. The level of detail for the 3D model was commensurate to a scale of representation 1:1000. Observation greed spacing was set at the following resolution or higher:X spacing = 30 mY spacing = 30 m

If point clouds of any kind or 3D digital surveys are not available for this scope, it is possible to use cartography. Digital cartography usually contains elements already defined in three dimensions, such as spot levels and contour lines. If they are not in 3D, they can be shifted vertically at their Z value to transform the 2D CAD cartography into a 3D CAD file, containing points and curves. If spot levels and contour lines are taken from a 3D digital survey or cartography, they should be collected into a file (dwg, txt or csv) as points or curves. We suggest using a CAD file, since this allows for instantaneously comparison of the result of the terrain against the source data in the same project view. The 3D CAD file is eventually linked to the BIM model by sharing coordinates. If a .txt or .csv format file is used, it does not need to be linked to the BIM project file but only chosen as external source data.

#### 2.3.3. Maps Themes

This research is based on the association between spatial and thematic data. The spatial data specifies the location and dimensions of a building in the maps, while the thematic data provides a description of a building’s characteristics through pre-set markers for the implementation of flood risk assessment and management.

Thematic maps allow us to visualise and sanitise complex data. The multilayer analytical approach we present follows a bottom-up information process; from the general information on the morphology of the territory, to the priority level to assign to each building for action during a flood. Below is a description of maps that should be made for adequate risk analysis.

**Existing-Terrain Map.** The DTM is the element from which other maps are created. Making building data and topographic data compatible and unified is, among many other things, one of the concepts of City Information Modeling (CIM), and it allows a smart building to be embedded in its territory—a required decision support tool for smart city planners. The DTM must be able to be visualised as pure terrain in a blanket map.

**Minimum Units Map**. MU Maps group all the units within the project scope area. A Minimum Unit (MU) represents a building or a portion of a building whose body and shape is configured as a minimum typological unit, with precise geometric and volumetric characteristics, and with a specific structural configuration. MUs are the elements used in the thematic maps for the attribution of risk factors and for risk management. All the MUs must have a unique ID code and their projected surfaces must be measurable (m^2^) since the measurements are needed for the estimation and attribution of funds in case of restoration post-damage.

The shape of the MU can be extracted from the city cartography as building footprint polylines provided on a CAD file, linked to the BIM model. The suggested scale of representation for the cartography is 1:1000. For smaller urban centres, more detailed representation scales such as 1:500 or 1:200 can be used, but that would drastically increase the time for the generation of the MU.

The MUs are created as subregions of the topography and, depending on the size of the city, can take many days to produce manually. Our research developed an algorithm to assist in this process by automatically drawing the buildings’ footprints as a subregion on BIM topographic surfaces.

Before designing a routine in any Visual Programming Language (VPL) compatible with BIM software, it is necessary to define a strategy and synthetically structure the pipeline of the algorithm (Figure 4) in order to write each of the intermediate tasks of the process. In this case, code was needed to read the CAD file with the building’s footprint, recognise the lines marking its perimeters, and draw on the topographic surface subregions for each building. Initial preparation in CAD consisted of drawing the buildings’ perimeters as closed polylines, and cleaning and purging the file from the unused object. The preparation in BIM consisted of modeling topographic surfaces and drawing model lines that would represent the wanted boundaries for the area in which we wished to run the routine. The CAD file, topographic surface, and boundary lines were selected as inputs for the VPL routine. The polylines in the CAD file were transferred to the VPL software and converted into compatible polycurves. Then, boundary lines were used to create an auxiliary solid used to delimit the algorithm’s area of operation. After constructing the solid, an intersection check was made between it and the polycurves to filter and delete the curves that were not within the chosen limits. With the filtered list, a custom node written in the Phyton language was needed to select all the polycurves and draw the subregions individually and automatically on the topographic surface in BIM. This algorithm is compact and can be reused in other situations, thus saving time in urban planning studies with BIM tools. The VPL file is available to the public for free access on the Zenodo platform with doi:10.5281/zenodo.7090669, and requires Dynamo Revit 2.3.

**Exposure Map.** The exposure map takes into account metric values of altitude and distance from the shore for each building included in the analysed area. The distance from the water is easy to check since it can be measured on a site plan. The altitude can be managed using reference levels at the values required by the analysis of the exposure, and defined in Section 2.1.2 as 4, 6, 8,10, and 12m. Floor plans at each level are generated automatically and allow for the visualisation of the submerged portion of the city for each scenario (Figure 5).

**Vulnerability Map.** This map groups building on the basis of the five levels of vulnerability defined in Section 2.1.3 of this article.

**Safe Risk Map.** The safe risk map is the result of the scores collected by each building in the Exposure and Vulnerability Map. It shows which buildings should be made safer by improving the buildings themselves or their surrounding areas.

The next three maps are used to define filters for the last map, the priority map, which is the one that should guide the management of post-flood rehabilitation.

**Ground Floor Use Map.** This map shows the functions of the building at street level.

**Basement Map.** This identifies the building with lower-ground or basement floors. A basement that is flooded for too long can compromise the building’s foundation and stability. The building can collapse, becoming a hazard to the public on the street and to neighboring buildings.

**Value Map**. This map identifies valuable buildings in the urban territory. It can be used as a reference to be consulted in case of repairs after catastrophic events, for the allocation of funds for reconstruction, and for planning reconstruction and restoration activities. The recognition of buildings of historical-architectural interest and their classification should be consistent with the attribution of significance of other existing plans, or any instruments suitable for preserving those buildings recognised as being of major interest. We defined five categories of value.

Value 1: This category groups those buildings of historical-architectural interest at a national or international level, i.e., buildings listed as ‘cultural heritage’ by national and international regulations, or buildings that serve as historic landmarks in the urban territory.Value 2: This category groups historic buildings, not included in the category above, that stand out for their high artistic or historical value. This category may include buildings of public interest, or groups of buildings that have an important significance for the identity of the community and should be preserved intact for future generations.Value 3: This category groups buildings that have characteristics attributable to the urban landscape of the city, or that are repeated to such an extent that they are easily recognisable and contribute to the aesthetic definition of the city itself.Value 4: Common buildings.Value 5: Building with no aesthetic, historical or social value including ruins or abandoned buildings.

Other indicators can be added to the proposed analysis according to local economic, cultural, social, or historic factors, and can be promoted or demoted to lower levels of value if necessary.

**Priority Map.** The priority map should be used after the emergency phase ends, which means after having saved as many lives as possible and moved people to safety. It is used to coordinate flood damage restoration operations. The purpose of this map is to bring people back to a pre-flood condition, however possible this may be. For this reason, residential buildings should be prioritised to be restored from damage (at the highest priority level).

High-exposure and high-vulnerability values are clearly associated with high levels of priority—the most exposed and vulnerable assets are taken care of in first stages, as Priority 1, 2, etc. Buildings with basements are also prioritised, since they are more prone to flood risk—with a consequent emphasis on emergency, transport, and rescue buildings with underground floors. Since the most urgent action during an emergency is providing safety to people, the Value categories in the Priority Map are shifted to lower Priority grades (Value 1 to Priority 3, Value 2 to Priority 4 and Value 3 to Priority 5). The exceptions in the priority map depend on the importance of the buildings in times of risk. After COVID-19, it has been recognised how important it is to have operative hospitals during cases of a catastrophe. That is why we give hospitals the highest priority. Transport stations play a role as interchange hubs for escape and can also serve as supply routes. Although hospitals, rescue and emergency assets, and transport stations are low-ranking within the Value category, they are all Priority 1. This is how priorities can be organised:Priority 1: Exposure 5, hospitals, transport, rescue and emergency.Priority 2: Exposure 4, Basement, Residence, Public building, Ruins (Securing buildings procedure) and Vulnerability 5.Priority 3: Exposure 3, Service, industry, office, commerce, Value 1, and others from Vulnerability 4 if not stated in a higher category.Priority 4: Exposure 2, Value 2, Social importance, and others from Vulnerability 3 if not stated in a higher category.Priority 5: Exposure 1, Value 3, and others from Vulnerability 2 if not stated in a higher category.

Markers can be added to the proposed analysis or moved to a more appropriate level according to local economic, cultural, social, or historic factors to be prioritised.

#### 2.3.4. Model Management

All the thematic maps listed above must coexist in the same project model. Their visualisation, as well as switching between them, must be quick and easy.

Project construction phases in BIM allow one unique terrain model as a base layer for all the maps. The main characteristics of the toposurface are that it is made to be highly adaptable to the evolution of the site, as well as its possible construction and demolition phases over time. It can be split into two or more parts, in subregions, as well as manipulated to create underground areas to allocate the lower ground and basement building levels.

Each map must be considered as a project scenario. For the eight maps, eight scenarios must be set. A scenario is a construction phase, and a one-to-one correspondence must be set between maps and phases; namely, the “existing” map belongs to the “existing” phase, the “minimum unit” map belongs to the “minimum unit” phase, and so on. It must be taken into account that phases have a chronological sequence. To avoid ambiguity, it must be guaranteed that one map per time is displayed for each construction phase. To do that, a demolition phase must be attributed to each map. If the generic map n is built in phase n, its demolition is set in the successive phase, namely phase n+1 (consider n the generic number that would represent the position of a map/phase in a complete list of all the maps/phases of the project that respects the chronological sequence of project phasing). Once each map is set within its proper construction and demolition phases, it is necessary to apply a visualisation filter that hides the demolition phase in all the project views. Once that is done, it is possible to switch between maps instantly, simply by selecting any phase listed in the project phase-menu.

## 3. Results

This section deals with the application of the methodology described in the previous sections on the case study, i.e., Lisbon city centre. We selected the area of Lisbon that lies between the city and the water, connecting the estuary of the Tagus River with the Atlantic Ocean, focusing on the urban area between the historic tower of Belém, located on the shore of the river, and Parque das Naçôes. The area hosts many buildings of various value, from Jerónimos Monastery (a UNESCO World Heritage Site), to governmental facilities, hospitals, and residential areas.

Our aim was to produce a model that would work as an instrument for gathering, analysing, and managing the information for risk management.

### 3.1. Dataset and Tools

#### 3.1.1. Terrain Model

A morphometric BIM model was created primarily from two sources (references can be found at the bottom of the article as Appendix A):An altimetric cartography in dwg format, scale 1:1000.A bathymetric map of the estuary of the Tagus River in jpeg format (spatial resolution 100 m).

We modeled the topography of the city of Lisbon from the altimetry map, while we used the bathymetric map to model the estuary of the Tagus River. We used the 2020 version of Autodesk Revit software to explore the application of BIM for Urban Planning. We exploited the characteristics of the “topography” BIM object and the construction “phases” to develop the thematic maps onto the toposurface, within a unique BIM project file. Google Maps and its “street view” function helped to verify each property, as well as the property’s status, current use, and value.

The terrain model was made on the basis of the altimetric cartography of Lisbon in CAD format. It did not share the same datum of the one used by the control points acquired by GPS along the riverside. A preliminary transformation of the CAD datum to the GPS datum was performed. Since we could not run the translation singularly for all the observations, we calculated an average translation vector and applied the same transformation to all the observations along x,y coordinates. The WebTransCoord of “Direção Geral do Território” provides an online tool for coordinate transformation, listing different kinds of standardized data as inputs and outputs. Our output datum is Datum73/Hayford-Gauss.

The cartography was provided in dwg format in squares of 800 m × 500 m rectangles. We obtained the coordinates of the corners of the rectangular CAD data, inserted them into WebTransCoord to calculate the transformed coordinates, then computed the translation vectors between homologous points. We finally computed the average of the translation vector. This last vector corresponds to the translation to apply to the dwg files. The error, intrinsic to this translation operation is small and negligible for the purpose of this research.

The CAD file was then linked to the model by share the same coordinate system.

The CAD file that merged all the 800 m × 500 m rectangles of the scope area had approximately 240 layers, not all of which were used to generate the topography. As a first step, we eliminated all objects other than points or polylines. Layers that included the collimation of points on surfaces other than the terrain were also eliminated, e.g., points identifying building heights, landmarks, street furniture, monuments, trees, and utility poles, among other objects. The file before being imported into Revit, only had curves and points at the ground level (Figure 6).

The vertical distance between points in the cartography was 1 m, while the distance on x,y axes was variable between ~3 and ~70 m. This is due to the presence of buildings, or rather blocks that cover large areas. The area near the riverbank also has a low density of points and the detail of the terrain is not rich.

The automatically generated toposurface on the 3D contours and points imported from CAD showed some imperfections near the coastline due to the way triangulations were automatically calculated in the most peripheral area. Once the toposurface was refined, we made sure to have it on the project phase “existing”, as it represented a land survey at the contemporaneous state (Figure 7).

The terrain modeling process was automatic and took only a few minutes. The terrain covers a 25 km long area of the north bank of the river and the file size is 95 MB.

#### 3.1.2. File Size

Due to the extension of the scope area, we had to evaluate a strategy for managing such a big model and make sure we could finalize our experiment. We wanted to make sure we could run the automatic generation of the terrain all in one go to avoid defects in the toposurface. In general, the RAM of the computer used determines the maximum usable file size. It is recommended not to create Revit files larger than one twentieth of the size of the RAM of the computer being used. Thus, if the computer has 16 GB of RAM, it is preferable to keep the file under 800 MB. However, in our experience, we do not recommend creating files larger than 250 MB, to ensure that the file is accessible from less powerful computers, and, above all, not to make operations too slow. It is better to address file sizes early in the design process than wait until the files are problematic, not leaving space for project growth.

At this stage, we decided to work with a smaller sample, an area of 5.5 km^2^, including the entire city centre of Lisbon, from the 25th of April Bridge to Santa Apolonia train station, in order to test the thematic maps and keep the size of the project file under control.

#### 3.1.3. Minimum Units

On the same project phase of the toposurface, namely the “existing” phase, we added the riverbed and the waterway (Figure 8). Next, we divided the toposurface into subregions for the identification of Minimum Units (MU). A subregion defines an area of the toposurface where different sets of properties, such as material, can be applied (Figure 9).

To each subregion, Revit provides values of the subsurface area and the subsurface projected area. It also assigns a unique identifier code that can be used to link the MU to internal elements or external data, such as spreadsheets, scripts, or plug-ins for compatible simulations. A block can have one or several MUs. Usually, their contours matched the property division and door numbers that we found in the CAD cartography. We checked the MU distribution using Google Street View since its 360 views were recently updated, in August 2021.

The operation for the creation of the subregions was time-consuming, so we created a routine in Dynamo Revit that used the pipeline described in the methodology section (Figure 4). The Dynamo Revit routine automatically creates subregions and requires to select only three inputs on the Revit model:a CAD file containing the polylines. This must be already linked to the Revit model.a topography on which the subregions must be created.model lines (a square, closed polygon, circle, etc.) that highlights the area into which the subregions must be created.

It is advisable to test the performance of a computer using an area smaller of 500 × 500 m and then try for bigger areas. We successfully reiterated this routine on squares of a pre-set grid of 800 × 800 m, on a workstation with the following specs:Processor: AMD Ryzer 7 4700G with Radeo Graphics 3.60 GHzRAM: 16.0 GMSystem type 64-bitAutodesk Revit version: 2020 (minimum version needed)Dynamo Revit version: 2.3.0.5885Topography projected area: 55 km^2^

Dynamo Revit turned Figure 10a into Figure 10b

#### 3.1.4. 3D Maps

Following the instruction of the Methodology section, after producing the MU identification map, we produced the other maps required for risk assessment (Figure 11). Those that deserved extra consideration, because of the choice of new markers, were the Value Map and the Priority Map. The markers for the two maps were adapted to the case study. We focused on the needs of the city of Lisbon as one of the oldest European capitals, with built heritage that is inestimable. We considered that the city in recent years has undergone social and economic transformation. It has gathered people coming from all over the world for its touristic attractions and been awarded Green Capital 2020 Lisbon, as pioneer in Europe for green infrastructure planning and sustainable mobility. For this reason, we added extra markers in the classification of the assets in the Value and Priority maps, namely: historic/touristic importance, proximity to the centre, and social importance/usage.

#### 3.1.5. Value Map

Common values were defined and identified to identify the significance and characteristics that distinguish the historic centre in order to create a charter of values that included an assessment of the architectural, landscape and social value of the buildings included in the scope of the research. According to the PDM (Plano Diretor Municipal) of Lisbon city, the municipal heritage structure includes immovable cultural assets of predominantly architectural, historical and landscape interest, which, due to their particular relevance, must be specially treated and preserved within the scope of management and planning acts. Our research takes into account the “Carta Municipal do Património Edificado e Paisagístico” from the PDM that includes immovable properties of national interest, public interest or municipal interest, from Saint George Castle to the shops of historical and/or artistic reference.

The value map defines a hierarchy of the buildings by their value, starting from their function and their recognized value. It helps for future interventions on assets belonging to the municipal heritage structure that must give priority to their long-term conservation and enhancement to ensure their identity and avoid their destruction and/or deterioration.

#### 3.1.6. Priority Map

The Priority Map is a synthesis map that defines the intrinsic value of each building based on parameters previously identified and contained in the other maps, supplemented also by considerations of each building’s position in the urban context of the historic centre, its social and economic value, and especially its importance in emergencies due to flooding.

The maps of values and priorities respect the classes of attribution described at Section 2.3.3. Priority 1 means high priority and includes “Hospital” and building with “Rescue and Emergency” service functions, Priority 5 means low priority and includes the markers “Historic importance “and “Proximity to the centre”. The scale was translated into colours to be visually represented on 3D maps. In Table 1 we compare some of the markers used for the case study and how their hierarchy position in the normal condition (Value hierarchy) changes after flooding (Priority Hierarchy).

The markers listed above are taken from the maps we made. Not all the markers from all the maps are shown in the table, only the ones that are considered exemptions for the attribution of the priority class, since we addressed this possible topic in the methodology section.

Figure 12 shows the scale of transformation of importance of building categories, from the ordinary “Value” rating (blue line) during normal times, into “Priority” rating (orange line) after cases of flooding.

A historic building in an ordinary situation has greater architectural, historical and social significance than a hospital. However, in the event of a natural disaster, the functioning of hospitals is crucial while that of the historic building is not. Therefore, the restoration of historic buildings should only be planned after the emergency phase is declared over and the socio-economic functions of the city are restored.

#### 3.1.7. Phases

The exploited feature for the creation of several maps in BIM was Revit’s construction “phases”. This choice made possible to operate on the same object, avoiding duplications—of the topography model in this case—minimising risk of creating too large a file.

In parallel with the definition of the construction ‘phase’ of each thematic map, it was also necessary to define the ‘demolition phase’ for each of them. Demolition defines the moment at which a given phase ceases to be displayed in a project view.

By default, Revit offers two project phases, “existing” and “new construction”. For this project, the existing phase hosted the toposurface of the city of Lisbon of the area within the extension of the scope of this research, two slab objects used for modeling the riverbed and the water body, and a third flat slab used to represent the south bank in the Almada council, which was not part of this study. The river model follows the levels on the bathymetric curves of the map of the estuary of the Tagus provided by the “Instituto Hidrográfico” of the Portuguese Navy. The other eight phases match the one proposed in the methodology section and host their homolog 3D map (Table 2).

BIM software provides a certain number of pre-set filters for the visualisation of the construction phases but none of them fitted our need to avoid the simultaneous visualisation of construction phase and demolition phase in the project view. To exclude the demolition phase from the visualisation phase we had to create a customised filter from the phase management panel. For a faster and more natural management of maps visualisation, we included each map in a new workset that could be turned on and off from the workset panel. Worksets operate like CAD layers. This extra feature of controlling the visualization of the maps from the workset panel avoided the risk of the maps being unintentionally moved in the wrong phase.

## 4. Discussion

The proposed model information system can provide a comprehensive and valuable tool for analysing flood events, forecasting targets, and planning strategies. In a spatial information system, collecting, processing and managing data are among the most important aspects, so the model must be adequately structured and easy to use. We tested the efficiency of the proposed methodology, implementing a preliminary analysis on the city centre of Lisbon using thematic 3D maps. We noticed that, in addition to map management through the demolition phases, exploring the model could be made easier for non-experts by creating worksets for each phase.

After our experiment on a large file such as Lisbon city centre, we can now say that risk assessment and risk management can be implemented in BIM using only an in-built tool.

For small centres, the file size management would be easier, and we consider that for territories up to a reference value of 15 km^2^ it is not necessary to split the model into sub-areas. However, if the extension of a city exceeds the reference value, we suggest creating a federated CIM model. A federated model uses a series of models linked to each other. It means that the BIM manager should plan in advance a structure on how the model should be federated. We suggest creating an empty model with all the phases, and replicating the grid that defines the areas that each sub-models contains. The sub-models should not exceed the reference file size. Federating the BIM model, also allows the project to be integrated with more information from other sources, for example, with 3D models of buildings, and enables interoperability with other stakeholders. A track table should be included as the front-sheet of the main model and must be updated with every new upgrade version of the model. The track sheet should include the number of its version (and keep a record of the previous ones), the general information of the project (name, number, location, company name), the structure of the model, the levels of detail and development attributed to the project or portion of project, the repository used for storing linked files and models and their location, the level of detail used for each part of the model, partial reviews of the model, name of the operators that worked at the project, and quality assurance responsible. A second sheet to be included in the BIM model is that showing Figure 2 and risk assessments matrices.

## 5. Conclusions

The flexibility of the BIM toposurface is a key point for this research and also for future updates of the model. The resolution used on the toposurface is connected to the scale of representation of the CAD altimetry from which it is generated, namely 1:1000. There are some areas in which the density of the measured points is low and can be improved, especially in the area close to the riverbank. In addition, the terrain model does not present walls and roadside decorations, which are elements that can be included in the future. A higher degree of accuracy for the toposurface is attainable just by manipulating it, by adding a major number of control points on the base of a point cloud. This will not affect the 3D maps or their phasing structure.

Improvements may be required if the model is used for the management of damage after floods. In the survey of damaged sites, the need to employ the most advanced surveying techniques is becoming increasingly evident, such as the use of terrestrial LIDAR (light detection and ranging) and/or low-altitude aerial photogrammetric survey systems, which can produce highly reliable three-dimensional maps and models. The latest and best remote sensing solution for the quick acquisition of point clouds for projects of large scale, such as that presented in this paper, is an MMS (Mobile Mapping System) based on SLAM (Simultaneous Localization and Mapping). It has proven more appropriate than laser scanning and drone imaging since it can dramatically reduce data acquisition and post-production times. Moreover, MMSs do not have to deal with all the privacy and permit issues that drone methods face. All the mentioned advantages of a mobile survey are at the expense of the quality of the point cloud, which, in this case, has higher noise values than a traditional laser scanner survey. However, the quality of data acquired by mobile scanners is sufficient for the requirements of urban surveys.

The use of the above-mentioned technologies also constitutes an effective database for the future monitoring of building complexes and historic centres.

In-depth knowledge of urban structures in terms of assessing architectural and environmental value, as well as knowledge of flood damage, is essential for reconstruction planning and design.

This research did not take into account the application of depth-damage curves, flood depth, or probability-damage curves, since an ongoing study conducted by the “Climate Change Impacts Adaptation and Modeling” research group on the Baixa area of Lisbon has been working on these. However, the results of the work [33] performed by “Climate Change Impacts Adaptation and Modeling” research group, if converted into thematic maps, could eventually be integrated with our model.

## Figures and Tables

**Figure 1 sensors-22-07456-f001:**
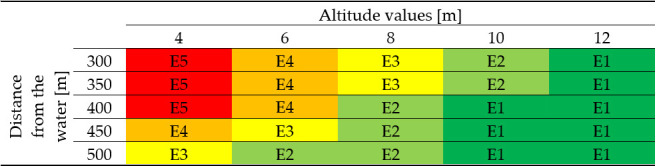
Risk exposure matrix. Level E1 (dark green); E2 (light green); E3 (yellow); E4 (orange); and E5 (red) distributed according to the two factors of exposure: distance from the water and altitude.

**Figure 2 sensors-22-07456-f002:**
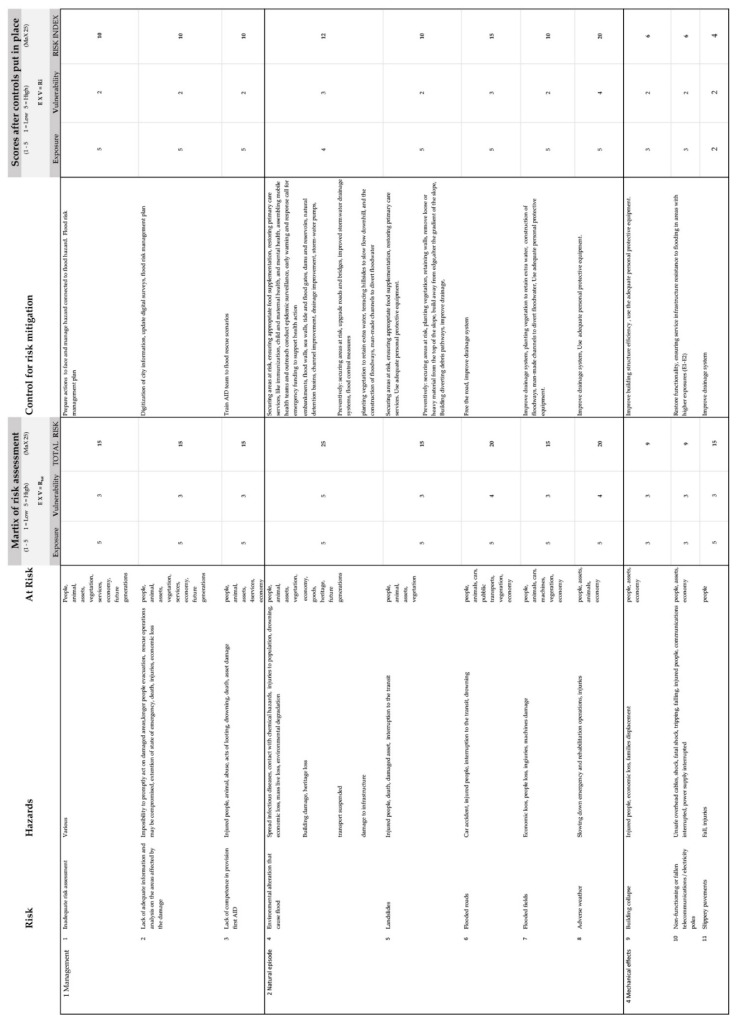
Risk assessment scheme.

**Figure 3 sensors-22-07456-f003:**
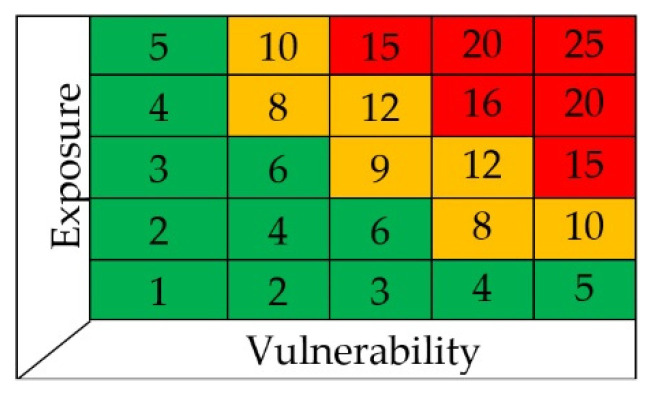
Safety risk acceptance matrix. Level 1 to 5 acceptable; 6 to 12 tolerable, but requires risk mitigation; 13 to 25 not acceptable.

**Figure 4 sensors-22-07456-f004:**
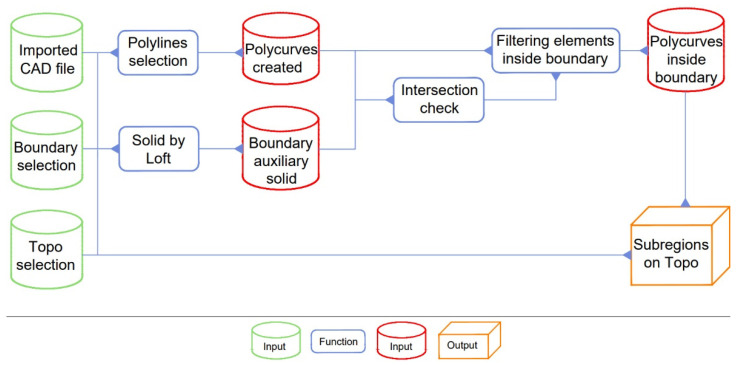
Pipeline for the VPL routine.

**Figure 5 sensors-22-07456-f005:**
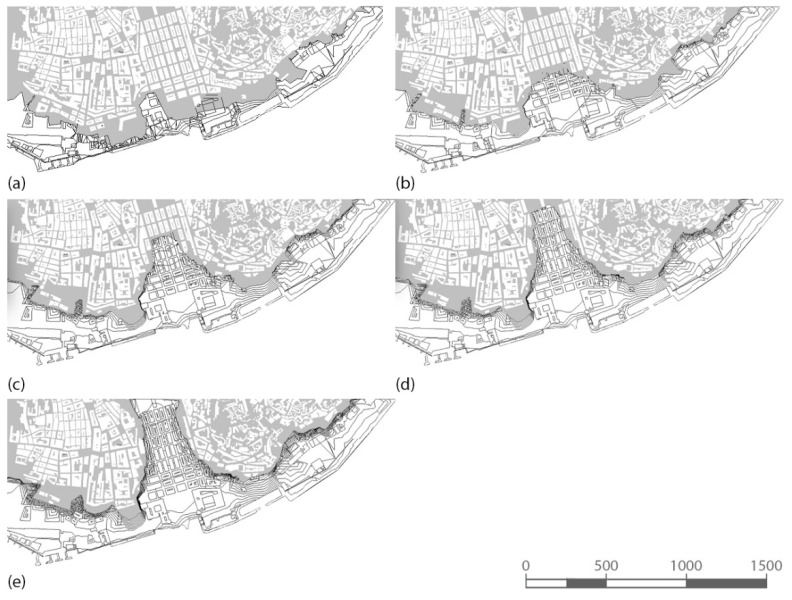
Exposure at different altitudes. The image shows, in grey, the position of the reference plane set at the following altitudes: (**a**) 4 m; (**b**) 6 m, (**c**) 8 m; (**d**) 10 m; and (**e**) 12 m.

**Figure 6 sensors-22-07456-f006:**
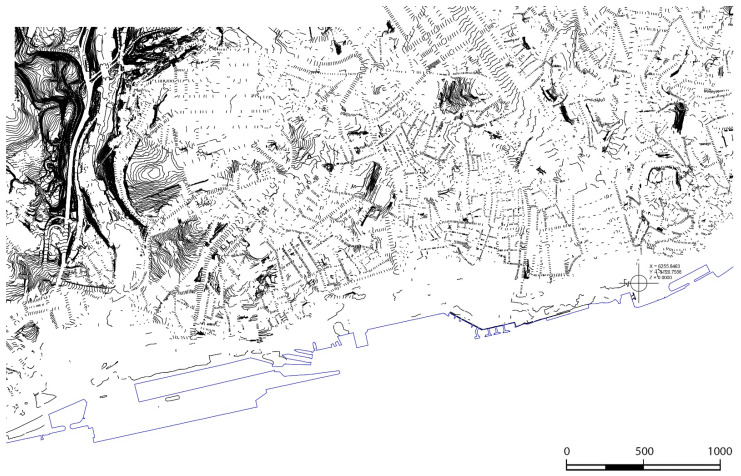
Contours on CAD, imported into Revit and used for the automatic creation of the terrain.

**Figure 7 sensors-22-07456-f007:**
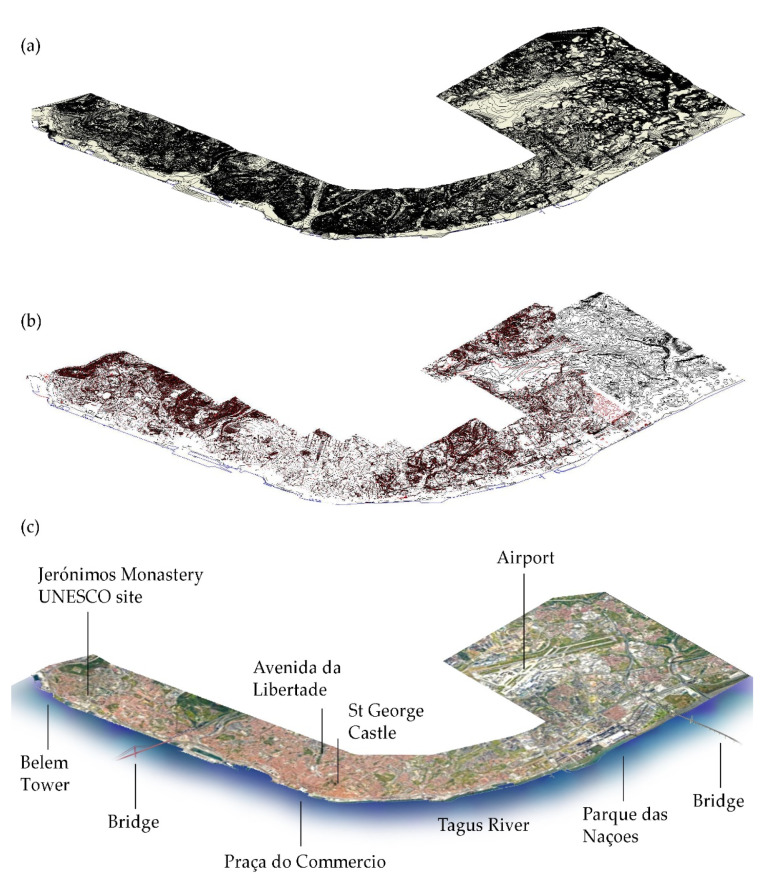
Scope of the research area. A CAD file with the altimetry of the city of Lisbon was scrubbed of the unneeded layers, purged and linked to the BIM file, and subsequently used for the automatic generation of the toposurface. (**a**) The 3D terrain modelled in Autodesk Revit in all its extension. (**b**) Curves and point observations taken from the cartography 1:1000. (**c**) Satellite view of the research area in which are highlighted some important spots as visual reference in the Lisbon council.

**Figure 8 sensors-22-07456-f008:**
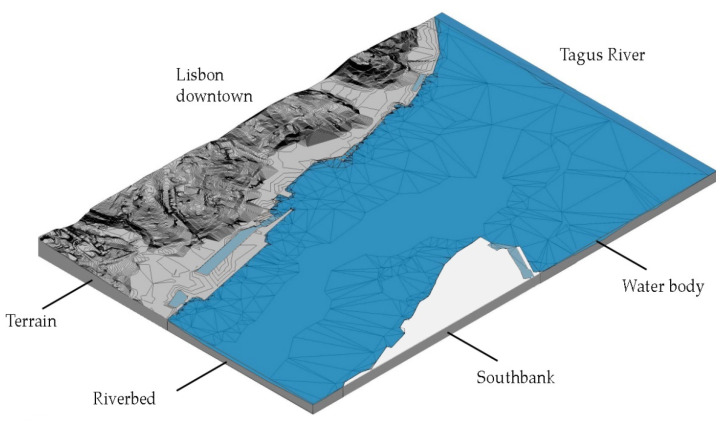
Detail of the model in proximity of the centre of Lisbon, from the 25th of April Bridge to Santa Apolonia train station. The main elements that compose the model shown are the toposurface that shapes the city, the two floors that form the riverbed and the waterway of River Tagus, and the flat floor used to represent the Southbank.

**Figure 9 sensors-22-07456-f009:**
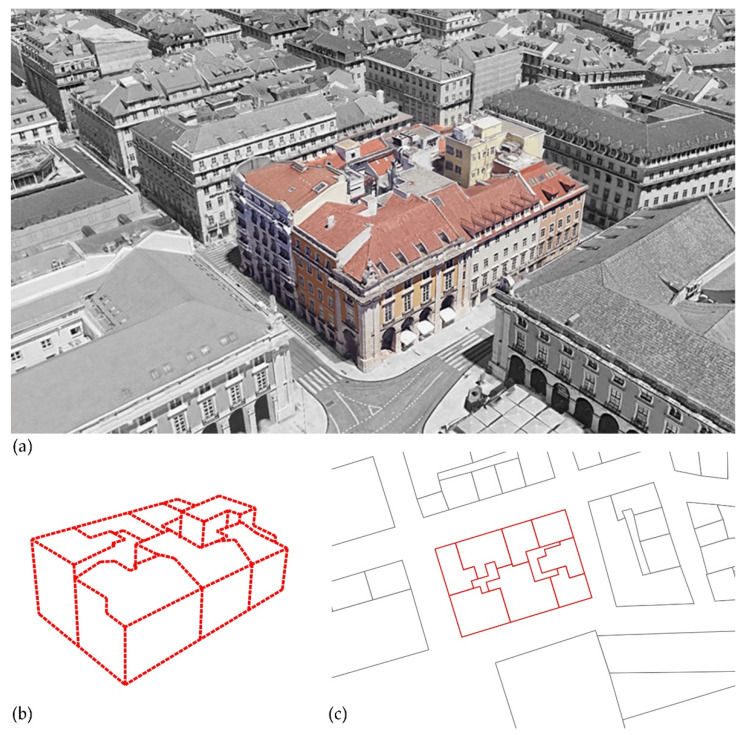
A block of the city (**a**) can present one or more MUs (**b**). They usually match with the division of the block in the cartography (**c**).

**Figure 10 sensors-22-07456-f010:**
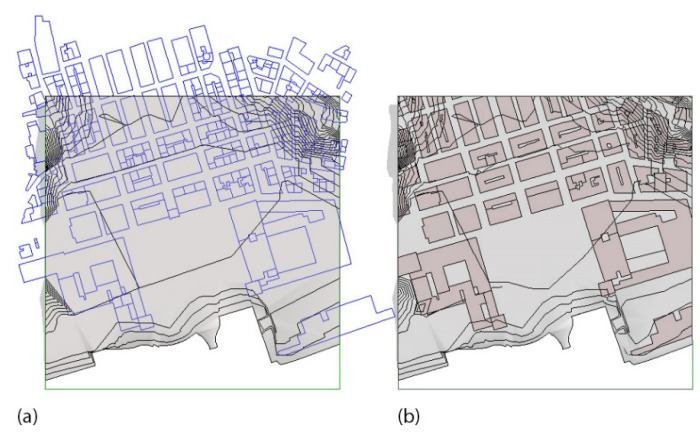
A portion of the topography model in a square of 500 × 500 m. Figure 10 (**a**) shows the input elements required by our routine for the automatic generation of subsurfaces: a CAD file for the polylines to be processer for the creation of the sub regions, the topography surface and the model lines (in green). Figure 10 (**b**) shows the results of the sub-regions on the toposurface.

**Figure 11 sensors-22-07456-f011:**
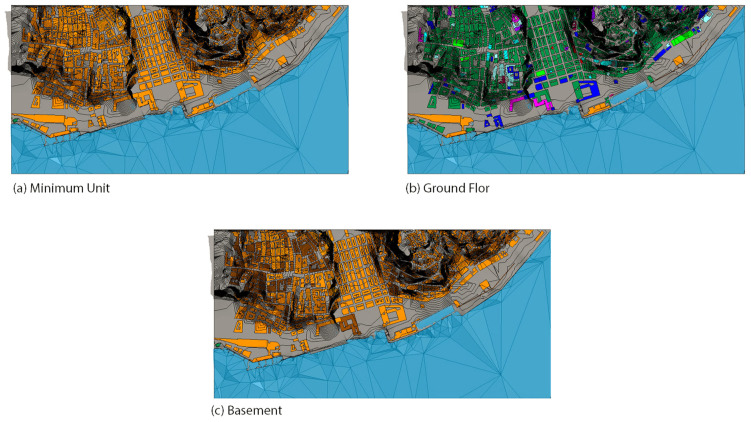
All the maps produced for this case study.

**Figure 12 sensors-22-07456-f012:**
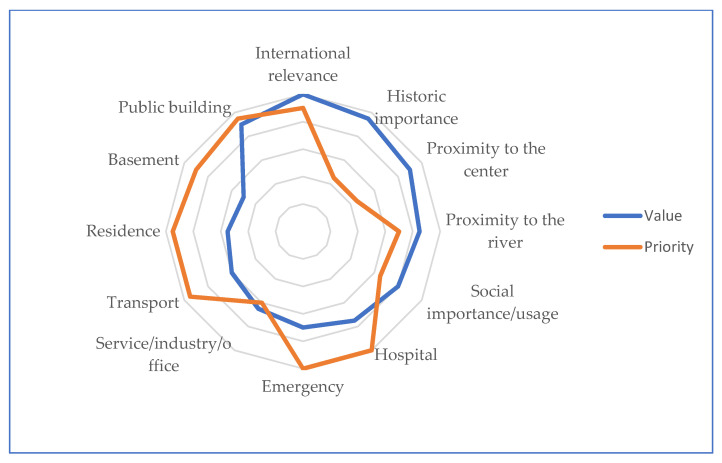
Comparison of the assigned degrees of value and priority for the principal categories of MU.

**Table 1 sensors-22-07456-t001:** List of 12 markers following the order for the “Building Value Hierarchy” and the “Building Priority Hierarchy”.

Scale	Building Value Hierarchy	Building Priority Hierarchy
1	International relevance	Hospital
2	Public building	Rescue and emergency
3	Historic importance	Transport
4	Proximity to the centre	Residence
5	Proximity to the river	Public building
6	Social importance/usage	Basement
7	Hospital	International relevance
8	Rescue and emergency	Proximity to the river
9	Service/industry/office	Social importance/usage
10	Transport	Service/industry/office
11	Residence	Historic/touristic importance
12	Basement	Proximity to the centre

**Table 2 sensors-22-07456-t002:** Visualization grid of the 3D Maps using the project phases. In red is marked the phase in which a 3D object or a map will start to be displayed and known as a construction phase. In yellow are the demolition phases, namely when they stop being visualized in the project view. In grey are the elements that will always appear in the project view, regardless of the chosen phase.

3D Element	Existing	MU	Exposure	Vulnerability	Risk	Ground F.	Basement	Value	Priority	Empty
DTM										
River body	On									
Bathymetry										
Minimum Units Map		On	Off							
Risk exposure Map			On	Off						
Risk Vulnerability Map				On	Off					
Safe risk Map					On	Off				
Ground floor Map						On	Off			
Basement Map							On	Off		
Value Map								On	Off	
Priority Map									On	Off

## Data Availability

Bathymetric map of the Tagus River is available on the website of the “Instituto Hidrográfico” at https://www.hidrografico.pt/op/33 (accessed on 1 July 2022), Cartography of the altimetry of Lisbon, scale 1:1000, is available on the website of the “Centro de Cartografia” of the Faculty of Architecture of the University of Lisbon at http://home.fa.utl.pt/www.fa.utl.pt/administracao/cartografia/sites.html (accessed on 1 July 2022); Dynamo Revit file available at: https://doi.org/10.5281/zenodo.7090669 (accessed on 1 September 2022).

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
