# Peer review of "A Preliminary Contribution towards a Risk-Based Model for Flood Management Planning Using BIM: A Case Study of Lisbon"

_sensors, 2022, doi:10.3390/s22197456_

Round 1
Reviewer 1 Report
The research presented is interesting. Tools and methodologies that exploit the use of BIM for the management of cities are currently required to increase knowledge in the areas of facility management, scenario simulation, links with the territories and their risks, with a view to the construction of digital twins, not only of buildings but also of cities (with all the complexities that this entails).
However, the article has several critical aspects that must be improved to publish.
General comments
While the thematic contribution of the research is relevant, the text has several aspects that need to be improved to become a scientific publication. Generally, the article's writing does not correspond to academic writing. The article comprises "loose sentences" poorly integrated into the story, adding aspects that are sometimes difficult to connect. In addition, the authors incorporate paragraphs on various topics, which are not entirely connected or related. In addition, the authors have contents that do not correspond to the respective chapters or seem to belong to other chapters. Sections are missing and/or others should be re-structured. In addition, the MDPI format should be followed.
Introduction
- The introduction section should be improved. The contents should be delivered in an orderly way, connected and grouped in paragraphs or groups of paragraphs, ranging from the general to the particular.
- The authors should clarify the problem to be solved, the research questions and/or the research objectives.
- The authors describe the application case in this section. This content is usually expressed in the Development, Application Case, Case Study, or Methodology sections (only if applicable). General aspects of it may be added in the introduction to point towards the research objectives or the problems to be solved, not as a complete description of the case itself. All the more so when the authors intend to present a "general method applied to the particular case of Lisbon", not to a specific case altogether.
A Background section could be added to address the problems and state of the art in more detail. The article, as it stands, lacks this information.
Materials and Methods
This section should be "neutral", showing the steps and details necessary for the implementation of the method. Instead, the authors mix a general description with the case applied to the study area. It is recommended to rewrite this section. Many of the elements mentioned can be carried over to the Results section.
This reviewer would expect the authors to deliver all recommendations, proposals, workflows, uses and roles for flood management using BIM. The scientific contribution of this type of article lies precisely in these aspects: to deliver a method or case study at a high level of detail so that other authors or interested parties can replicate it in different contexts.
Results
This section needs to be considerably improved. The authors address four sections as results: 3D maps, Value Map, Priority Map, and Phases. In none of the sections is the full background of the case study, which allows for an understanding of how the case study was developed. While the authors show results and some of the criteria used, these can and should be improved considerably.
Therefore, several questions need to be answered throughout this section:
- What details and types of information are contained in the BIM model? At what level of detail should the model be built? What level of interoperability does the model have? What tools were used to make the model? was any visual programming used, or could it be used to improve the efficiency of the development of this type of model?
- What is the proposed flood management tool or method, how does this methodology or tool work in practice, and what are the challenges, roles and usability aspects that need to be considered?
- In addition, all details of hardware and software used should be indicated. In this type of model (city management), the weight of the generated files is a major problem. Often the amount of information incorporated makes the models difficult to use (or requires computers with higher capacities), is this the case?
It would be interesting to see graphically how Table 1 and Figure 6 translate into different management or effects on the model created.
The authors indicate the use of Revit's "phases" tool. It is important that these aspects are mentioned in a general way (at a conceptual and working level) and then materialised in specific software. Beyond that (which is an aspect that should also be addressed in the methodology), the authors should indicate and show the use of this tool graphically (although this review prefers to call it: a way of managing the models). The information provided gives few relevant aspects of the Revit tool itself, but what is interesting is its use and exploration at the management level that the authors propose in the article.
Discussion
While the authors address interesting aspects (and not discussed in the article), which this reviewer appreciates, it is expected that the authors first discuss their method and results and then talk about options to improve their proposal. The authors should discuss the practical opportunities of the technique, the usability and the improvement it represents over the way of handling traditionally addressed issues.
Conclusions
This section should be added.
Author Response
Thank you reviewer for your feedback. At the moment of our first submission the article was not as we wanted it to be, but we did our best to not miss the opportunity of publishing and be forced to decline the invitation. We rushed to write. We are sorry if you had to deal with unclear parts.
We have improved the contents and the article structure following your suggestions.
Thanks again
The research team.
----------------------------------------------------------------------------
Review: The research presented is interesting. Tools and methodologies that exploit the use of BIM for the management of cities are currently required to increase knowledge in the areas of facility management, scenario simulation, links with the territories and their risks, with a view to the construction of digital twins, not only of buildings but also of cities (with all the complexities that this entails).
However, the article has several critical aspects that must be improved to publish.
Authors: We have improved the article with the contents requested
General comments
Review: While the thematic contribution of the research is relevant, the text has several aspects that need to be improved to become a scientific publication. Generally, the article's writing does not correspond to academic writing. The article comprises "loose sentences" poorly integrated into the story, adding aspects that are sometimes difficult to connect. In addition, the authors incorporate paragraphs on various topics, which are not entirely connected or related. In addition, the authors have contents that do not correspond to the respective chapters or seem to belong to other chapters. Sections are missing and/or others should be re-structured. In addition, the MDPI format should be followed.
Authors: Improvements have been applied to the structure and contents of the article, as suggested.
Introduction
Review- The introduction section should be improved. The contents should be delivered in an orderly way, connected and grouped in paragraphs or groups of paragraphs, ranging from the general to the particular.
Authors: We have reorganized and improved the introduction
Review- The authors should clarify the problem to be solved, the research questions and/or the research objectives.
Authors: Introduction should now include the information required
Review- The authors describe the application case in this section. This content is usually expressed in the Development, Application Case, Case Study, or Methodology sections (only if applicable). General aspects of it may be added in the introduction to point towards the research objectives or the problems to be solved, not as a complete description of the case itself. All the more so when the authors intend to present a "general method applied to the particular case of Lisbon", not to a specific case altogether.
Authors: We moved the description of the case study to appropriate section
Review A Background section could be added to address the problems and state of the art in more detail. The article, as it stands, lacks this information.
Authors: Background information added
Materials and Methods
Review: This section should be "neutral", showing the steps and details necessary for the implementation of the method. Instead, the authors mix a general description with the case applied to the study area. It is recommended to rewrite this section. Many of the elements mentioned can be carried over to the Results section.
Authors: This section has been rewritten following reviewer’s suggestions
Review: This reviewer would expect the authors to deliver all recommendations, proposals, workflows, uses and roles for flood management using BIM. The scientific contribution of this type of article lies precisely in these aspects: to deliver a method or case study at a high level of detail so that other authors or interested parties can replicate it in different contexts.
Authors: This section has been rewritten following reviewer’s suggestions
Results
Review: This section needs to be considerably improved. The authors address four sections as results: 3D maps, Value Map, Priority Map, and Phases. In none of the sections is the full background of the case study, which allows for an understanding of how the case study was developed. While the authors show results and some of the criteria used, these can and should be improved considerably.
Authors: Criteria used are reported now
Therefore, several questions need to be answered throughout this section:
Review: - What details and types of information are contained in the BIM model? At what level of detail should the model be built? What level of interoperability does the model have? What tools were used to make the model? was any visual programming used, or could it be used to improve the efficiency of the development of this type of model?
Authors: Specifications added
Review: - What is the proposed flood management tool or method, how does this methodology or tool work in practice, and what are the challenges, roles and usability aspects that need to be considered?
Authors: Methodology section now contains this information
Review: - In addition, all details of hardware and software used should be indicated. In this type of model (city management), the weight of the generated files is a major problem. Often the amount of information incorporated makes the models difficult to use (or requires computers with higher capacities), is this the case?
Authors: Information about the file size are considered in the article
Review: It would be interesting to see graphically how Table 1 and Figure 6 translate into different management or effects on the model created.
Authors: the connection between the list of markers in table 1 and figure 6 is not relevant for this research.
Review: The authors indicate the use of Revit's "phases" tool. It is important that these aspects are mentioned in a general way (at a conceptual and working level) and then materialised in specific software. Beyond that (which is an aspect that should also be addressed in the methodology), the authors should indicate and show the use of this tool graphically (although this review prefers to call it: a way of managing the models). The information provided gives few relevant aspects of the Revit tool itself, but what is interesting is its use and exploration at the management level that the authors propose in the article.
Authors: “ Maps theme”, “ Map and model management” paragraphs added to the Methodology section
Discussion
Review: While the authors address interesting aspects (and not discussed in the article), which this reviewer appreciates, it is expected that the authors first discuss their method and results and then talk about options to improve their proposal. The authors should discuss the practical opportunities of the technique, the usability and the improvement it represents over the way of handling traditionally addressed issues.
Authors: this paragraph has been changed
Conclusions
Review: This section should be added.
Authors: Section added
Reviewer 2 Report
The article is well structured and the idea of integrating BIM and GIS information into an unique system for the management of flood emergencies is very interesting. It represents the first step of further results, and in the future the integration of other dataset could improve the research (roads network, quality of terrain, flood probability, etc). Well done!
Below there are few corrections to the manuscript:
line 193- please insert "area" instead of "are";
line 265 - please insert all the thematic maps in a list, or in a table,
to be described better ad more clearly explained.
line 288 - please insert "defines" instead of "define".
line 392 - please write "dramatically reduce".
Author Response
Thank you reviewer for your feedback. At the moment of our first submission the article was not as we wanted it to be, but we did our best to not miss the opportunity of publishing and be forced to decline the invitation. We rushed to write. We are sorry if you had to deal with unclear parts.
We have improved the contents and the article structure following your suggestions.
Thanks again
The research team.
----------------------------------------------------------------------------
Review: The article is well structured and the idea of integrating BIM and GIS information into an unique system for the management of flood emergencies is very interesting. It represents the first step of further results, and in the future the integration of other dataset could improve the research (roads network, quality of terrain, flood probability, etc). Well done!
Authors: Thank you very much. The article has been improved.
Below there are few corrections to the manuscript:
line 193- please insert "area" instead of "are"; Done
line 265 - please insert all the thematic maps in a list, or in a table,
to be described better ad more clearly explained.
Article has been improved in contents and structure
line 288 - please insert "defines" instead of "define". Done
line 392 - please write "dramatically reduce". Done
Reviewer 3 Report
As it is presented in the abstract, row 24 “we have developed risk-based 3D maps” there is no such map presented in the article, just some complementary thematic maps are presented. Also a risk map would involve some hydraulic/hidrological calculation, taking in calculation also the average annual flows.
Line 77: the collaboration between BIM and GIS still lacks a unified standard for information exchange, creating ambiguity
It would be recommended to give some details about the way the ambiguity is created and maybe enumerating some main factors that should be standardized so that BIM and GIS could be more interoperable.
Lines 94-95: European and national directives in the construction field are pushing for BIM to become the 95 standard, especially for publicly funded construction works
It would be suggested to specify the directive number, law etc, as the legislation and regulations might change over the time.
Lines 127-128: The BIM model would work as a unique instrument for gathering, analysing, monitoring and extracting the information for the risk management
How would you use the BIM to perform analyses for risk management, what software solutions would you use? is it REVIT capable of risk management analyses? Please give some more details.
Line 139: An altimetric cartography in dwg format, scale 1:1000
Please specify the source for the altimetric data. Is it vectorised from raster, is it from a local survey? What is the main projection used for this data?
Line 140: A bathymetric map of the estuary of the Tagus River in jpeg format (spatial resolution 100 m)
Please specify the projection for this bathymetric data. Also it would be suggested to specify how did you integrate the raster data with the vector data and also how did the spatial resolution of the raster and the accuracy of the vector data presented before would change the output results of the risk management.
Lines 157-158: we calculated an average translation vector and applied the same transformation to all the observations along x,y coordinates
Did you checked the results? Adding some more control points, others than those used for the average would be suggested. Did you compute the error for the transformation?
Line 270: Figure 5. Use of the Ground Floor 3D Map
Adding a legend and a scale bar for the map would add value to it.
Lines 391-392: It has proven more appropriate than laser scanning and drone imaging since it can dramatically data acquisition and post production times
Please rephrase the phrase…… ex: “dramatically data acquisition”?
Author Response
Thank you reviewer for your feedback. At the moment of our first submission the article was not as we wanted it to be, but we did our best to not miss the opportunity of publishing and be forced to decline the invitation. We rushed to write. We are sorry if you had to deal with unclear parts.
We have improved the contents and the article structure following your suggestions.
Thanks again
The research team.
-------------------------------------------------------------
As it is presented in the abstract, row 24 “we have developed risk-based 3D maps” there is no such map presented in the article, just some complementary thematic maps are presented. Also a risk map would involve some hydraulic/hidrological calculation, taking in calculation also the average annual flows.
Authors: Methodology section has changed. Maps added to the article
Line 77: the collaboration between BIM and GIS still lacks a unified standard for information exchange, creating ambiguity
It would be recommended to give some details about the way the ambiguity is created and maybe enumerating some main factors that should be standardized so that BIM and GIS could be more interoperable.
Authors: We add reference about this topic
Lines 94-95: European and national directives in the construction field are pushing for BIM to become the 95 standard, especially for publicly funded construction works
It would be suggested to specify the directive number, law etc, as the legislation and regulations might change over the time.
Authors: Reference added
Lines 127-128: The BIM model would work as a unique instrument for gathering, analysing, monitoring and extracting the information for the risk management
How would you use the BIM to perform analyses for risk management, what software solutions would you use? is it REVIT capable of risk management analyses? Please give some more details.
Authors: Detail added in the Methodology section
Line 139: An altimetric cartography in dwg format, scale 1:1000
Please specify the source for the altimetric data. Is it vectorised from raster, is it from a local survey? What is the main projection used for this data?
Authors: Reference about CAD file at the bottom of the article
Line 140: A bathymetric map of the estuary of the Tagus River in jpeg format (spatial resolution 100 m)
Please specify the projection for this bathymetric data. Also it would be suggested to specify how did you integrate the raster data with the vector data and also how did the spatial resolution of the raster and the accuracy of the vector data presented before would change the output results of the risk management.
Authors: All the in-depth information about our sources, bathymetric data included, can be consulted at theirs source. References are included at the bottom of the article.
Lines 157-158: we calculated an average translation vector and applied the same transformation to all the observations along x,y coordinates
Did you checked the results? Adding some more control points, others than those used for the average would be suggested. Did you compute the error for the transformation?
Authors: The error, intrinsic to this operation, is small and negligible for the purpose of this research. We add this description in the article.
Line 270: Figure 5. Use of the Ground Floor 3D Map
Adding a legend and a scale bar for the map would add value to it.
Authors:That map was shown in 3D view. Anyway, we replaced that figure with a new one.
Lines 391-392: It has proven more appropriate than laser scanning and drone imaging since it can dramatically data acquisition and post production times
Please rephrase the phrase…… ex: “dramatically data acquisition”?
Authors: Rephrased
Round 2
Reviewer 1 Report
The authors have improved the article considerably. The recommendations of this reviewer have been considered and implemented.
It is recommended that the following points be improved:
- Table 1 should go to Annex.
- The formatting should be improved, in line with MDPI guidelines.
Reviewer 3 Report
The manuscript was significantly improved.